# User Sentiment Analysis Based on Securities Application Elements

**DOI:** 10.3390/bs14090814

**Published:** 2024-09-13

**Authors:** Minji Kim, Subeen Kim, Yoonha Park, Sangwoo Bahn, Sung Hee Ahn, Bhavadharani NambiNarayanan

**Affiliations:** 1Department of Artificial Intelligence, Kyung Hee University, Yongin 17104, Republic of Korea; minjikim2000@khu.ac.kr; 2Department of Industrial and Management Systems Engineering, Kyung Hee University, Yongin 17104, Republic of Korea; sbkim8519@gmail.com (S.K.); pyhmaria@gmail.com (Y.P.); bhavamek20@gmail.com (B.N.); 3Department of Bigdata and Management Engineering, Namseoul University, Cheonan 31020, Republic of Korea

**Keywords:** aspect-based sentiment analysis, user review, securities application

## Abstract

Designing securities applications for mobile devices is challenging due to their inherent complexity, necessitating improvement through the analysis of online reviews. However, research applying deep learning techniques to the sentiment analysis of Korean text remains limited. This study explores the use of Aspect-Based Sentiment Analysis (ABSA) as an effective alternative to traditional user research methods for securities application design. By analyzing large volumes of text-based user review data of Korean securities applications, the study identifies critical elements like “update”, “screen”, “chart”, “login”, “access”, “authentication”, “account”, and “transaction”, revealing nuanced user sentiments through techniques such as PMI, SVD, and Word2Vec. ABSA offers deeper insights compared to overall ratings, uncovering hidden areas of dissatisfaction despite positive biases in reviews. This research demonstrates the scalability and cost-effectiveness of ABSA in mobile-application design research.

## 1. Introduction

Security applications provide consumers with enhanced convenience by offering real-time protection against cyber threats, ensuring safe online transactions, and securing personal information. These applications simplify complex security tasks, such as managing passwords, monitoring potential breaches, and authenticating user identities, thereby allowing consumers to confidently perform daily activities without extensive technical knowledge. Streamlined interfaces contribute to a seamless user experience, reducing the cognitive load associated with maintaining digital security. However, due to the inherent complexity of these tasks and the extensive range of features and menus, enhancing usability and achieving high levels of user satisfaction remain challenging. Therefore, it is crucial for designers of security applications to investigate issues that diminish user satisfaction in order to develop more effective and user-friendly solutions.

In Korea, where mobile usage is particularly high, financial transactions, including securities trading, are commonly conducted via mobile applications, similar to other services. As the mobile era rapidly progresses, enabling users to access information and carry out nearly all financial transactions without the constraints of time and space, the demand for security applications has surged. Previous studies have confirmed that user satisfaction with security applications positively influences the continued usage intention [1,2]. Consequently, there is an emerging need to enhance user satisfaction by providing high-quality security application services while continuously updating the systems to reflect rapidly changing user characteristics and environments. Therefore, it is essential to design user-centered mobile trading systems that fully incorporate the unique features of smartphones as a medium and align with consumer needs. Despite this need, however, research applying deep learning techniques to sentiment analysis of Korean text, particularly in the context of security applications, remains limited [3].

In order to obtain direct and specific information related to the customer needs for user-centered design, focus group interviews, user observation, user surveys, and card sorting are commonly used by researchers [4]. However, in order to derive meaningful results from a large sample, these methods require a lot of manpower, cost, and face-to-face interaction with subjects. In addition, user research and experiments using the above methods are often not feasible due to the shift towards social distancing—a result of the impact of COVID-19. Also, when dealing with products that are in the early stages of growth during the product life cycle, it is difficult to recruit a large number of users who have experienced the product. Smartphone applications, which have become essential tools for modern people, exhibit diverse characteristics even when used within the same domain, much like products at different stages of growth. As a result, selecting research subjects from a globally dispersed user base is challenging. This is where text mining for user analysis can be more effective. In fact, the growing amount of information online has led to the rise of text analytics as an alternative to traditional research methods. There is a lot of research going on in various fields, and there are already examples of real-world applications [5].

Online reviews are a useful resource for mobile-application designers as they provide a valuable source of feedback and insights from users who have used the application. These reviews can aid developers in understanding the aspects of the application that users find appealing or dislike, as well as areas that need improvement. This feedback can help identify common issues that users encounter, assist in evaluating the user experience, and allow developers to assess user preferences and expectations. The benefits of utilizing online reviews for mobile-application design are numerous. First, identifying common user issues through online reviews can alert developers to any recurring issues that require immediate attention. Also understanding user preferences and expectations from online reviews can help designers create more user-friendly applications that are tailored to users’ needs. Additionally, evaluating the user experience based on online reviews can provide insight into areas of the application that need improvement or further development, and positive reviews can help improve the overall rating of the application, making it more appealing to potential users. Therefore, it can be concluded that online reviews are a valuable resource for mobile-application designers as they provide a wealth of information and feedback that can be used to enhance the user experience and improve the overall quality of the application.

When utilizing text analytics across platforms like the web, SNS, and apps, it offers the advantage of quickly identifying and responding to real-time user opinions that frequently change due to factors like application updates, based on the analysis of user review data on app platforms. Through this study, we attempt to establish an analysis methodology for the user experience, extracted based on text mining, and devise a method that can be meaningful for data analysis and is comparable to other existing analysis methods.

The underlying framework utilized in this study for user experience analysis is Aspect-Based Sentiment Analysis (ABSA), which aims to obtain opinions on detailed objects (Aspects) and features of a product within the review text written by product users. This is information that cannot be obtained from user ratings or general review-level sentiment analysis. By extracting informative words from a large number of user product reviews that contain information about specific objects and interaction between users and products, ABSA can obtain product elements (Aspects) and derive the user sentiment for each element. ABSA has been applied to analyze text data related to restaurants [6,7,8], smartphone apps [9], education [10], trips [11], etc.

In this study, we assumed that analyzing users’ sentiments towards each specific aspect or feature of a product, rather than relying solely on the overall product rating, would provide a more diverse and unbiased understanding of user opinions about the product. The rationale behind this assumption is the well-known online user-behavior tendency where typically only users who had a positive experience leave positive reviews and ratings. User reviews are usually skewed towards the positive side, forming a J-shaped distribution, with the long tail being dominated by positive reviews [12]. However, even when users leave such positively biased overall reviews, they may still not be fully satisfied with all aspects or features of the product. For instance, when using a messaging app like KakaoTalk, users may leave positive overall reviews and ratings for the general app experience, but they could have had negative sentiments towards specific components like the gifting feature or money transfer functionality, which are expressed in the review text. Therefore, to identify desired improvement areas and specific complaints, it is necessary to extract user sentiments for each distinct element or feature of the product embedded within the review text. Consequently, in this research, we focused on performing aspect-level sentiment analysis. Based on this, we attempted to design a sentiment analysis system that can comprehensively identify and correlate the factors influencing user sentiments and product performance, with the aim of providing meaningful insights to complement and potentially replace existing user research methods.

## 2. Literature Review

### 2.1. Sentiment Analysis

Sentiment analysis is a tool for analyzing unstructured data, examining materials that reveal individual tendencies, such as people’s words or writings, to extract emotional vocabulary on the topic and classify emotions [13]. Sentiment analysis is a type of text mining, also known as opinion mining, that analyzes whether the sentiment on the topic is positive, negative, or neutral, using product or service reviews, or opinions. It can also be used to create a sentiment dictionary that reveals the nature of the text. A typical sentiment analysis simply categorizes or predicts the sentiment of a given sentence or paragraph as positive or negative.

The contemporary digital landscape has witnessed a significant proliferation of “sentiments” that can be found online, including on popular social media platforms such as Twitter, Facebook, YouTube, and various blogs, as well as e-commerce platforms like Amazon, eBay, and Alibaba. These textual expressions are of great value to companies and individuals seeking to obtain prompt feedback on their products or services, assess their overall reputation, and adjust their actions accordingly. In this regard, sentiment analysis technologies enable these entities to monitor diverse social media channels in real time and make informed decisions based on the resulting insights. The direct beneficiaries of this technology encompass a wide range of stakeholders, including marketing managers, PR firms, campaign managers, politicians, equity investors, and online shoppers.

Sentiment analysis can be classified into several categories [14,15], including (1) document-level [16], (2) sentence-level [17], and (3) aspect-level [18] sentiment analysis. In document-level sentiment analysis, the entire document is analyzed as a single unit [19]. The main advantage of document-level sentiment analysis is that it can provide an overall opinion on the context of the document. However, this approach assumes that the review is written by a single person, which is a limitation. Both supervised and unsupervised classification algorithms are used in this method. Sentence-level sentiment analysis involves breaking down the document into individual sentences, which is called subjectivity classification. Each sentence is analyzed independently. The challenging aspect of this approach is that it requires identifying the target of each sentence, so if the target of the sentence is not known, determining the polarity (positive, negative, neutral) of the sentence is not useful [20]. Aspect-level sentiment analysis requires the feature terms of a product, which are the main target of the analysis [21]. This approach primarily depends on the attributes of the target entity. It has mainly been applied to analyzing reviews, feedback, comments, and complaints.

### 2.2. Aspect-Based Sentiment Analysis (ABSA)

Aspect-Based Sentiment Analysis (ABSA) is a natural language-processing technique and considers the terms associated with each subaspect and attribute of a target and analyzes the sentiment for each in detail. While typical sentiment analysis mostly assumes that there is only one aspect and one polarity in a given text, ABSA predicts the sentiment associated with each aspect and element mentioned in a sentence. Since each element is analyzed, it has great practical value in that it is easy to improve each element. Initial research using ABSA was based on the premise of the “frequent repetition of key vocabulary”, and the method of calculating sentiment polarity involved extracting nouns with a high frequency of occurrence as elements, but recently, the method of identifying aspect terms or aspect categories contained in the text and judging the sentiment of them has attracted attention.

ABSA has the potential to provide a useful tool for analyzing user surveys. ABSA is capable of identifying and analyzing the sentiment of specific aspects or features of a product or service, thereby offering a more detailed understanding of user feedback. For example, by analyzing the sentiment of various aspects of a product, such as its design, ease of use, reliability, and customer support, ABSA can help companies identify which aspects are most positively and negatively perceived by customers.

The benefits of using ABSA for analyzing user surveys are numerous. Firstly, ABSA provides a more granular understanding of user feedback, allowing decision-makers to identify specific strengths and weaknesses of a product or service. Secondly, ABSA can automate the process of sentiment analysis, making it faster and more efficient than manual analysis. Finally, ABSA can help decision-makers to make more informed decisions about product development, marketing, and customer support, by providing more detailed insights.

In summary, ABSA has the potential to provide a valuable tool for analyzing user surveys. By identifying sentiments about specific aspects or features of a product or service, ABSA can provide a more detailed understanding of user feedback, making it easier for decision-makers to identify areas for improvement and make informed decisions. In addition, there have been studies using ABSA in sentiment analysis. First of all, there was a study to obtain user opinions by extracting the parts and features that are the main elements of electric vehicles through user-review analysis based on ABSA using machine learning [22]. There is also a study that derived practical measures to increase the utilization value of BERT by using ABSA [23].

The ABSA method with SVM was applied to classify sentiments in reviews as positive, negative, or neutral, with performance evaluated using various textual features [24]. The experiments conducted on restaurant and laptop review datasets demonstrated the effective applicability of ABSA. ABSA was used to analyze mobile-application reviews, enabling the identification of user requirements that were then applied to enhance app development [25]. ABSA was also applied to smart government application-review data by integrating domain-specific lexicons and rule-based techniques to extract key elements and classify sentiments [26]. It was further shown that incorporating these lexicons and rules as input features in an SVM model led to higher accuracy.

However, in the case of other studies that used ABSA, most of them conducted opinion mining to derive sentiment scores for each element, and there is a regret that no further research was conducted. This study utilized partial techniques to compensate for these ABSA shortcomings.

## 3. Methods

### 3.1. Framework

The research framework of this paper is as shown in the Figure 1 below.

### 3.2. Word Embedding

In natural language processing and machine learning, embedding refers to the process of representing words or phrases in a numerical vector space, also known as a vector representation. This allows words and phrases to be processed as numerical inputs, making them more amenable to machine learning algorithms.

Embeddings are often created through the use of neural networks, specifically embedding models, which take a large corpus of text as input and generate a vector representation for each word in the corpus. These embeddings capture semantic and syntactic relationships between words, such that words that are semantically similar or appear in similar contexts are represented by vectors that are close together in the vector space.

These embeddings can be used as input to a wide variety of machine learning models for natural language-processing tasks such as text classification, sentiment analysis, and language translation. By using embeddings, natural language-processing models can more effectively process and understand human language, which is often complex, nuanced, and highly context-dependent. There are several techniques available for creating word embeddings—for example, frequency-based methods, prediction-based methods, etc.

TF-IDF is used as a representative technique of frequency-based embedding, and Word2Vec, employing CBOW or skip-gram, is used as a representative technique of prediction-based embedding.

#### 3.2.1. Term Frequency—Inverse Document Frequency (TF-IDF)

TF-IDF in text mining can be applied to determine the weight of each word in a document [27]. TF-IDF is a statistical measure of how important a word is in a particular document across multiple documents. TF represents the word frequency, which indicates how many times a word appears in a document, and IDF is the inverse document frequency, which indicates how common a word is across documents. In other words, the higher the frequency of a word, the more important it is, and it has the advantage of providing a more intuitive indication of important words.

#### 3.2.2. Word2Vec

Word2Vec is a word-embedding method and is a neural network-based algorithm. Word2Vec generates word embeddings by projecting the words in a text into a vector space. The Word2Vec model proposes two model architectures: continuous bag-of-words (CBOW) and skip-gram, where CBOW predicts the central word based on the surrounding words, and skip-gram predicts the surrounding words based on the central word. Since Word2Vec uses context to vectorize words, it has the advantage that words with similar contexts can be represented by similar vectors in the vector space. Building on these advantages, Word2Vec has been widely used for embeddings in ABSA [28,29,30].

### 3.3. Aspect Extraction

#### 3.3.1. TextRank

TextRank is a graph-based technique for finding quality sentences in a textual document by indicating how a particular word relates to other sentences. It is derived from PageRank, which replaces the concept of pages in PageRank with the concept of words. First, a word graph or sentence graph is built, and then the key words or sentences in the document are extracted in ranking order using PageRank, a graph ranking algorithm.
(1)$$WS\left(V_{i}\right)=\left(1-d\right)+d\sum_{V_{j}\in In(V_{i})}\frac{w_{ji}}{\sum_{V_{k}\in Out\left(V_{j}\right)}w_{jk}}WS(V_{j})$$
where *V* represents a vertex. *E* is an edge, G = (*V,E*) is a directed graph,$\;In(V_{i})$ is the set of vertices pointing to a point (predecessor), $Out\left(V_{j}\right)\;$ is the set of vertices pointing to a point (successor), and d is a damping factor, which is a value between 0 and 1.

#### 3.3.2. Point-Wise Mutual Information (PMI)

Point-wise mutual information (PMI) is a measure of the association between two random variables and is used as an indicator to identify the polarity of emotional words or sentences by analyzing the similarity between words [31]. PMI is used for examining binary co-occurrences, but it has been adapted to handle more complex, multivariate problems. For example, it can be applied to identify subject–verb–object triples in text, offering insights into intricate relationships among multiple variables [32]. The PMI index is calculated through the assumption that if the two words to be analyzed have similar semantic polarity, the probability of occurrence within the same document is high [33]. In this study, we first created unigrams and then used skip-grams to group related words. The PMI index was then derived from the clusters, and the PMI index was used to generate subwords and create boxes. The PMI’s sensitivity to marginal probabilities can lead to a preference for rare or less-frequent events, which can skew results. To mitigate this, adjustments are sometimes made to PMI calculations to account for these biases, allowing it to align more closely with other association measures when marginal effects are controlled.

In general, PMI is defined as follows:(2)$$PMI\left(w_{1},w_{2}\right)={log}_{2}\frac{p\left(w_{1},w_{2}\right)}{p\left(w_{1})p(w_{2}\right)}\;$$
where $w_{1}$: term1 and $w_{2}$: term2.

What this expression means is the co-occurrence probability of two terms, where the polarity of the target term is determined by the difference in relevance between the positive and negative terms.

In practical applications, PMI has proven useful beyond text analysis. For instance, in remote sensing, it plays a crucial role in the unsupervised segmentation of hyperspectral and LiDAR data. Here, PMI helps integrate spatial, spectral, and elevation information, enhancing the analysis and interpretation of complex datasets.

Although computing PMI in real-time streaming scenarios is challenging, due to the need for continual updates on co-occurrence statistics, advances in approximation algorithms have made it feasible. These algorithms provide accurate estimations even with extensive vocabularies, broadening PMI’s applicability across various domains and types of data.

### 3.4. Model Description for Aspect Sentiment Classification

#### 3.4.1. Logistic Regression

Cox [34] proposed a probability model to predict the probability of an event using a linear combination of independent variables. The linear combination of feature vectors and weights is put into a logistic-sigmoid function to obtain the probability that the feature vector belongs to a certain label. When the probability is the largest, it belongs to that label.

#### 3.4.2. Support Vector Machine (SVM)

SVMs are a class of machine learning algorithms for classification that use margins to improve generalization in classification. Non-linear SVMs utilize kernel tricks to map data from a low-dimensional space to a high-dimensional space. After finding the linear decision boundary in the high-dimensional space, it is mapped back to the low-dimensional space to find the non-linear decision boundary before proceeding with classification learning based on the decision boundary. In ABSA, the use of SVMs demonstrates higher accuracy and a shorter computation time compared to traditional neural network models [35].

#### 3.4.3. LSTM (Long Short-Term Memory)

A recurrent neural network (RNN) is an artificial neural network for learning data that changes over time, such as time-series data. These RNNs are models developed to transfer data information from one point in time to the next, stored in each hidden layer, and can take into account non-linear relationships between data as well as temporal information. However, in reality, RNNs are limited in considering the long-term dependency of data due to the problem of gradient decay in the learning process. A new model that has emerged to overcome this problem is LSTM (Long Short-Term Memory).

#### 3.4.4. One-Dimensional Convolutional Neural Networks (1D CNN)

Convolutional neural networks (CNNs) are artificial neural networks that are mainly used to learn the features of image data using a kernel. In a traditional convolutional neural network, the kernel moves in only one direction, called a one-dimensional convolutional neural network (1D CNN), which is often used for time-series data.

#### 3.4.5. Singular Value Decomposition (SVD)

SVD is a powerful matrix factorization technique with diverse applications. It decomposes a rectangular matrix into three simpler matrices: two orthogonal and one diagonal. SVD is robust to numerical errors and exposes a matrix’s geometric structure, making it valuable for various matrix computations. Its applications range from analyzing political positions to measuring crystal growth rates and examining quantum entanglement. SVD is particularly useful in determining matrix rank and approximating matrices of a given rank. It can also be employed to solve both consistent and inconsistent systems of linear equations by computing orthonormal bases for the four fundamental subspaces associated with a matrix. With the increase in computational power and multidimensional data availability, higher-dimensional generalizations of SVD have become increasingly important, spurring new research in theoretical and applied mathematics [36,37].

## 4. Experiment

### 4.1. Dataset

In this study, we selected “securities applications” as a detailed research target to conduct sentiment analysis through store review data. The prolonged COVID-19 crisis, which began in 2019, boosted the stock market, with the KOSPI surpassing the 3300 level and causing significant changes in the domestic and international economy [38]. As interest in and participation in the stock market continued to grow, most non-professionals making personal investments naturally used the trading functions of smartphone securities applications (MTS), which allow them to buy and sell stocks easily and anywhere, rather than trading through PCs (HTS), which are limited by time and space. However, compared to the number of users of securities applications, which has been growing steadily in this trend and will exceed 10 million by 2020, there are still many users who are dissatisfied with the inadequate UI/UX environment. Table 1 shows an overview of the dataset, and Table 2 and Table 3 present examples of the review data of MTS applications in Korean and English, respectively.

In this study, we use a supervised learning method to build a sentiment analysis model. Therefore, we required experimental data with reviews and ratings. Therefore, we collected data from “Google Play Store,” where Android application users download applications and write reviews. The subjects were MTS applications of five Korean securities companies: “Korea Investment”, “Mirae Asset Securities M-Stock”, “Kiwoom Securities Hero Moon”, “KB Securities” and “Namu Securities”. The review data were collected using “Google Play Scraper”, a library that can crawl data from the Google Play Store using Python.

### 4.2. Pre-Processing

The data used in this study are Korean review data, which require various steps of pre-processing for Korean natural language processing. In addition, since sentiment analysis in this study is a supervised learning-based analysis method, it is necessary to label each review (positive and negative). Therefore, we performed the following data preprocessing steps:

Step (1)Removal of duplicate and missing data.Step (2)Extraction of Korean data using regular expressions.Step (3)Tokenization of data based on Korean morphemes.Step (4)Removal of non-Korean words such as surveys.Step (5)Labelling of data based on ratings.

After removing duplicate and missing data, the total number of review data used for sentiment analysis was 44,781. Due to the small amount of data, we used OKT, the fastest Korean morphological tokenizer, for tokenization. Finally, we labelled the ratings of the review data on a 0–5 point scale—positive for scores above 3 and negative for scores below 3—to train the sentiment analysis model. The number of positive and negative data in the pre-processed data is shown in the figure below.

### 4.3. Word Embedding

Since computers do not recognize natural language, it is necessary to vectorize it through the process of embedding and then proceed with modeling. In the case of word embedding, words are vectorized to enable computation, and words with similar meanings are mapped to a vectorially close space to convert words into dense representations.

Before embedding, padding was performed to ensure the review data were in the same dimension by making each review the same length.

As shown in Figure 2, when analyzing the length of the reviews, we found that the average review was 15 words, and the maximum was 325 words. However, as can be seen from the graph, most of the review data were within 50 words, so we used a padding factor of 50. Padding was performed with zero padding and pre-padding using KERAS’s preprocessing tools.

After padding, we used TF-IDF and Word2Vec as embedding methods depending on whether the model was supervised or unsupervised. TF-IDF is a type of count-based embedding that uses the frequency of the word and the frequency of the inverse document to weigh the word. Word2Vec is a type of inference-based embedding that can preserve contextual meaning by placing words in a vector plane. Therefore, we applied both count-based embedding and inference-based embedding to embed the model before training.

### 4.4. Aspect Extraction

After the morphological analysis of all the review data, we utilized TextRank technology to identify which words within the reviews have significant importance. We extracted the top 50 most-important words from the TextRank results; excluded meaningless words such as de, get, and it; and categorized all the words. We divided them into three categories: elements, application-related elements, and consumer needs-related elements. We then re-ranked the importance of the element words based on how often they appeared in the review to identify the top eight most-important elements.

After categorizing the top eight factors, we added words related to the top factors using the PMI index and adopted about five words with a high PMI index for each factor as sub-factors.

### 4.5. Sentiment Analysis

For modeling, several supervised learning models were selected and used. As supervised learning models, we adopted logistic regression and SVM, which are most commonly used as classifiers, and LSTM and CNN, which are types of recurrent neural network. After training a total of four models with review data, one model with the highest performance was selected by utilizing the accuracy and F1-score, which can identify model performance. The adopted model was later utilized as a model for extracting the sentiment score by element. After dividing the review data into sentence units using the adopted model, sentences containing relevant elements were re-entered into the model to conduct the analysis.

#### 4.5.1. Extracting Sentiment Scores by Brokerage Firm

After extracting the sentiment scores, we returned to the original data for each securities company and obtained the sentiment scores for each element by securities company to understand which securities company was the strongest in each element compared to the other securities companies.

#### 4.5.2. Review Classification Using Word2Vec

Word2Vec is an embedding technique that maps words into a vector space by learning from the context of the sentence. Since the embedded words are located in a vector space, the similarity of the words can be seen through the similarity of the vectors. Using the vectors embedded by Word2Vec, we can obtain not only the similarity between words but also the similarity between words and sentences. Therefore, in this study, we estimated the similarity of each element extracted in the element-extraction step and the entire review sentence, and then we extracted the reviews with the highest similarity to analyze the cause of the sentiment analysis results in more detail. The procedure for analyzing review similarity using Word2Vec is as follows:

Generate a DTM (document-term matrix) using the one-hot-encoding method.Generate a weight matrix using the following equation so that words that are close to the query word have a high weight and words that are not have a low weight.Create a matrix that extracts only element words from the weight matrix, and then create a score for each element by fusing it with DTM.

#### 4.5.3. Review Classification Using PMI and SVD

As a final post-training exercise, we utilized PMI and SVD to extract the important reviews related to the elements and check what users consider important for each element. We used the PMI index to find highly relevant words and SVD to reduce the dimensionality of the vector by keeping the most important information in a large vector space. After finding the words with a high PMI index, we restored the original vector to identify which reviews had a high PMI. We extracted the top 10 reviews and analyzed the users’ sentiments in detail.

## 5. Results

### 5.1. Aspect Extraction Results

We checked the element-extraction results for the review data and found that we can extract the elements as shown in the figure above. Table 4 shows the results of categorizing the financial application-review data into five main components: update, screen, authentication, feature, and app, and categorizing the review keywords according to the components. The table shows the frequency of occurrence of the following keywords in the five aspects of the application reviews. We selected eight factors that had a relatively high frequency of occurrence and could influence application user sentiment. Using these, we applied sentiment analysis to the five application reviews.

### 5.2. Results of Sentiment Analysis

Table 5 presents the results of training candidate models for analysis using application-review data, measuring the accuracy and F1-score. After comparing four models, the LSTM model, exhibiting the highest performance, was adopted. The adopted model conducted sentence-level sentiment analysis by inputting sentences containing elements and extracted element-specific sentiment scores.

## 6. Discussion and Conclusions

The table below illustrates the results of element-specific sentiment analysis for each application review using Word2Vec and LSTM. “Update”, “Screen”, “Chart”, “Account”, “Transaction”, “Login”, “Access”, and “Authentication” were selected as elements in the previous element-extraction process and then applied to the review data.

Figure 3 represents the results of sentiment analysis conducted on five application-review datasets, categorized into five components. Overall, it is evident that for all components of all applications, the number of negative reviews surpassed the positive ones. Specifically, reviews regarding updates were the most prevalent, followed by those concerning the screen.

Figure 4 depicts the results of sentiment analysis represented as percentages. It shows the ratio of positive to negative sentiment in sentences containing each element in each application review. Overall, a 30:70 positive-to-negative ratio was observed, but instances of a 15:85 ratio, such as in the case of “Screen”, “Account”, and “Authentication” for KoreaInvest Securities, were also identified. However, there were no cases where the positive ratio exceeded 40% for all elements in all applications.

The Table 6 below presents the results of review classification using PMI, SVD, and Word2Vec.

PMI, SVD and Word2Vec methodologies reveal critical issues in various aspects of the application, even though they focus on slightly different aspects. PMI and SVD emphasize technical and functional problems such as loading, access, and authentication issues post-update, as well as complex registration processes and difficulties in recognizing stored certificates during transactions. In contrast, Word2Vec highlights user interface and usability concerns, focusing on post-update errors, UI design, and so on. Additionally, Word2Vec emphasizes errors during various authentication methods and the need for additional features during transactions. While PMI and SVD addressed broader technical and performance issues, Word2Vec provided a complementary perspective by focusing on specific UI and user experience shortcomings, together offering a comprehensive analysis of user reviews.

ABSA breaks down user feedback into specific aspects (e.g., update, screen, chart, sign-in, access, authentication, account, and transaction). This helps in identifying which features or elements users are mentioning, and their sentiments towards each. Instead of predictable and general feedback, ABSA helps us to identify specific pain points and positive highlights. For example, users may like an application but have issues with the update process or the authentication mechanism. By analyzing sentiments associated with different aspects, companies can prioritize which areas need immediate attention. Aspects with a high volume of negative sentiment can be flagged for urgent improvements. ABSA allows companies to allocate resources more efficiently by focusing on the aspects that significantly impact user satisfaction. This focused approach ensures that development efforts are aligned with user needs.

Detailed insights from ABSA enable businesses to enhance specific features that users find problematic. For instance, if users are not satisfied with the screen layout, developers will focus on improving the UI design first. Understanding sentiments towards different aspects helps in designing products that cater to user preferences and expectations, and this approach leads to a more user-centric product. Utilizing Aspect-Based Sentiment Analysis for user-needs analysis and VOC analysis provides a detailed understanding of user feedback. It helps in identifying specific weakness and strengths, prioritizing development efforts, enhancing customer experience, making informed strategic decisions, and ensuring continuous improvement. By focusing on the review sentiments in users’ language, companies can better meet user expectations and improve overall satisfaction.

While general sentiment analysis, topic modeling, and keyword extraction each have their strengths, ABSA stands out for its ability to provide detailed, aspect-specific sentiment insights. This makes ABSA particularly powerful for user-needs analysis and VOC analysis, where understanding the nuanced sentiments of users towards specific features is crucial. ABSA offers a level of granularity and actionable insight that other text-mining methods often lack, making it an essential tool for targeted product development and customer satisfaction improvement.

## 7. Conclusions

Research on sentiment analysis using deep learning for Korean text is scarce. The challenges may stem from the complexity of the Korean language, the language-agnostic nature of deep learning models, or the lack of appropriate datasets [39]. Given the importance of sentiment analysis and the progress in other languages, expanding research in this area in Korean is essential.

This study used review data to analyze the factors that influence user experience in brokerage applications, identifying elements such as “update”, “screen”, “chart”, “login”, “access”, “authentication”, “account”, and “transaction” as critical to users. Item-level sentiment analysis revealed users’ positive and negative sentiments, providing a nuanced understanding. Using PMI, SVD, and Word2Vec techniques, the study extracted significant reviews related to these elements and performed in-depth sentiment analysis. The integration of various text-mining and sentiment analysis techniques proved effective, with neural networks such as the LSTM model demonstrating high performance in sentiment classification. In addition, Aspect-Based Sentiment Analysis (ABSA) provided deeper insights compared to traditional methods by extracting sentiments about specific aspects of the text.

Compared to traditional user research techniques such as surveys and focus groups, ABSA offers a scalable and cost-effective approach that is particularly effective in the post-COVID-19 era. A key finding was the identification of positive bias in online reviews, where generally positive sentiments masked specific areas of dissatisfaction, such as updates, screen design, and authentication processes. These detailed insights are critical for developers to target for improvement. However, the study faced limitations such as data quality issues, difficulties in delineating application elements, and challenges in processing Korean-language text [40,41].

Future research should focus on refining data preprocessing, fine-tuning element analysis, and applying various tokenization techniques to Korean data to improve the robustness of sentiment analysis. The practical significance of this study is substantial, as ABSA enables companies to make informed strategic decisions in product development, marketing, and customer support, thereby providing a competitive advantage by meeting user expectations and improving overall satisfaction. Ethical considerations, such as ensuring user privacy and maintaining transparency in the use of feedback, are critical to maintaining user trust and must be addressed in sentiment analysis practices to comply with privacy regulations.

## Figures and Tables

**Figure 1 behavsci-14-00814-f001:**
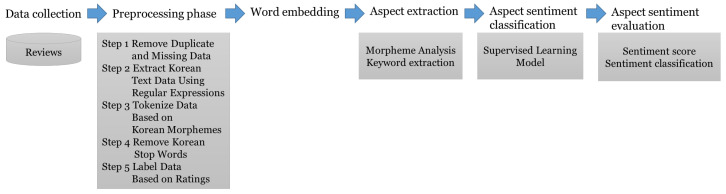
The framework of this study.

**Figure 2 behavsci-14-00814-f002:**
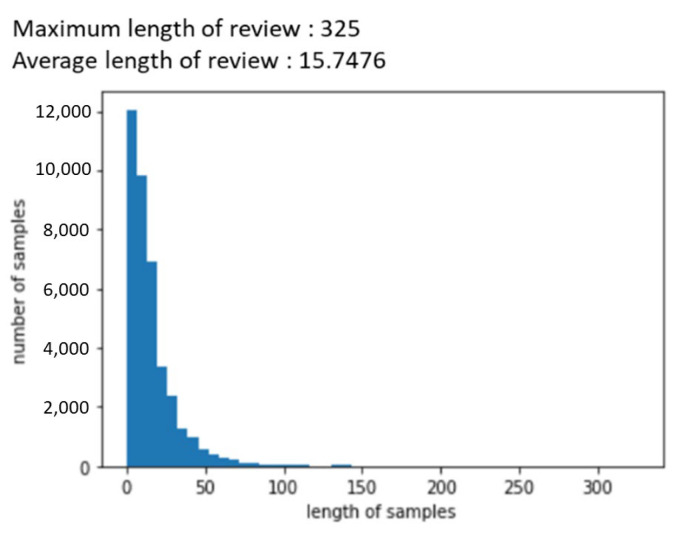
The length of reviews.

**Figure 3 behavsci-14-00814-f003:**
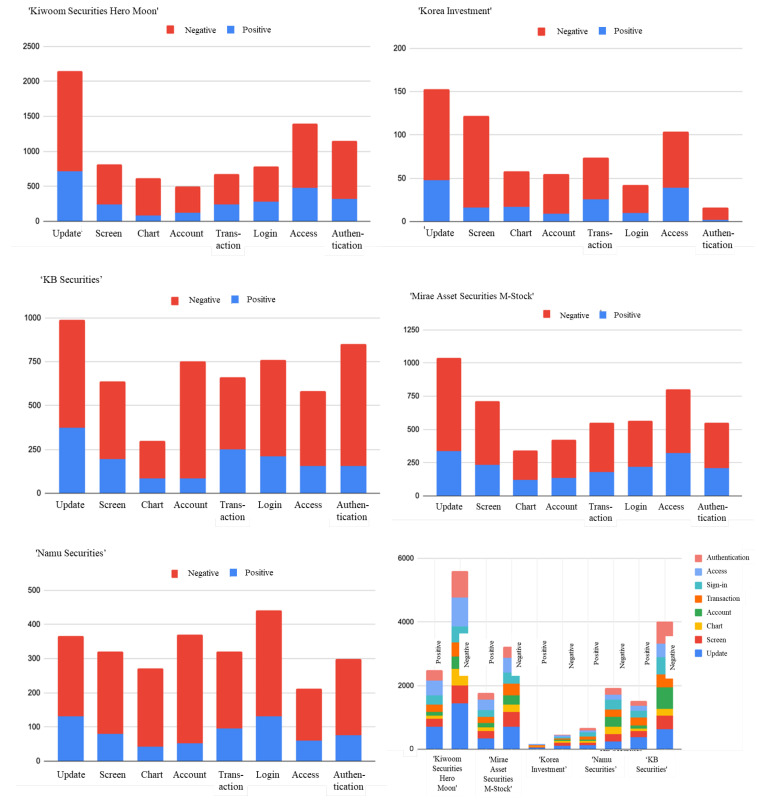
The results of sentiment analysis conducted on five application reviews.

**Figure 4 behavsci-14-00814-f004:**
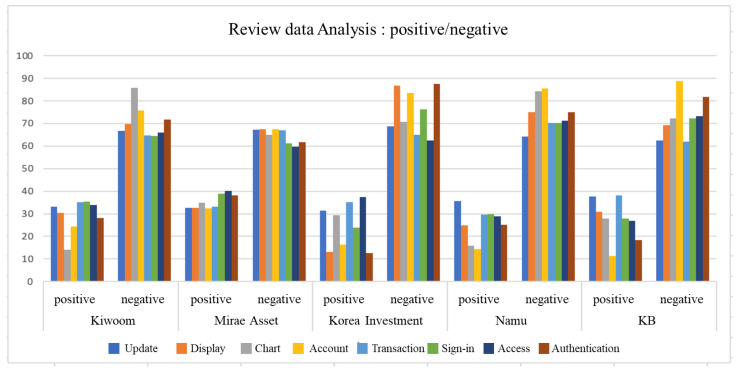
The results of sentiment analysis represented as percentages.

**Table 1 behavsci-14-00814-t001:** Overview of the review dataset of five Korean securities companies’ MTS applications.

MTS App Name	Number of Data	Period of Data Collection	Review Created Version (Range)	Language
Korea Investment	1412	June 2022~October 2022	1.01.04~1.01.17	Korean
Mirae Asset	9810	June 2017~October 2022	1.0.1~7.0.9	Korean
Kiwoom	17,968	January 2018~September 2022	0.9.0~5.5.9	Korean
KB	10,504	January 2016~November 2022	2.0.0~5.3.6	Korean
Namu	5111	October 2011~November 2022	8.45~8.93	Korean

**Table 2 behavsci-14-00814-t002:** Examples of the MTS applications’ review data (Korean).

MTS App Name	Score	Review Created Ver.	Review
Korea Investment	1	1.0.16	어제 업데이트 후 자꾸 팅기네요 어플 실행해서 거래를 할수가 없네요.
	2	1.01.16	메뉴 누를 때 최종 선택한 메뉴 위치좀 유지해 주세요 너무 불편해요 ㅠ
	3	1.01.17	기존에 앱에 있던 차트에서 ‘서랍종목’ 기능을 좀 추가해 주세요 매우 유용한 기능입니다
	4	1.01.14	적응하는데 좀 애먹었지만 종목요약 생긴것도 유용하고 나름 만족합니다~ 근데 전에는 ETF도 종목 토론방을 볼수 있었고 운용 보수율도 볼수 있었는데 사라졌네요ㅠㅠ 요것좀 개선해주세요~
	5	1.0.06	기존 앱보다 사용하기 편해졌습니다. 다크모드 기능이 추가되면 더 좋을 것 같습니다.
Mirae Asset	1	7.0.8	8년간 사용한 입장에서 후기 올립니다. 전 버전에 비해 속도가 너무 느려졌어요! 차트도 배너창 처럼 넣어놔서 직관성도 떨어집니다. 화면간 이동이 너무너무 느려요!!!!!!
	2	7.0.5	복잡하고 이전처럼 심플한게 좋은데 체결도 이전건 볼수도없고 아주 산만함
	3	7.0.3	우측에 피드가 계속 뜹니다. 아무리 찾아봐도 숨김 설정도 안되어 있고, 화면자체도 불편합니다. 이전에 정말 잘 사용하고 있었는데 업데이트후 불편하네요
	4	7.0.5	좀 더 발전된 모습을 보니 좋네요, 깔끔해진 것 같아요. 다만 다른 메뉴로 이동 시 렉이 걸리는 문제가 있으니 빠른 시간 내에 해결되었으면 좋겠습니다
	5	7.0.8	최고에요~~!! 디스플레이도 보기 편하고 세련된 디자인인 것 같습니다!! 또한 호가창과 주문창도 유저 프렌들리한 것 같습니다~!
Kiwoom	1	5.5.4	UI가 끔찍할 정도로 조악합니다. 메뉴 찾기 힘들고 조작하기 힘든 건 물론이고, 가장 기본적인 매수/매도주문조차 직관적이지 않은 부분이 많습니다.
	2	5.5.7	매수한 종목을 내 마음대로 정렬할수 있도록 해주셨으면 합니다. 자동정렬이 되니까 너~~~무 불편합니다
	3	5.2.5	기능은 좋다고 생각하는데 모바일로 증권하는 사람이 계속 화면만 보고 있지 않을텐데 다른 앱을 보거나 화면을 껐다 켜면 키움앱이 꺼져버려서 당황스러워요. 혹시 설정이 있는건지… 영웅문S, 영웅문G 모두 그러네요
	4	5.5.7	전 앱에 비해 깔끔해져서 보기 좋은건 있는데요 시간외 거래대금 상위 가 사라져서 조금 불편하네요 추가해주시면 좋을 것 같습니다
	5	4.9.5	# 위젯 만들어주심 좋을듯 # 모바일 화면구성시 좀 더 가독성 좋은 화면구성요망 (예. 관심종목에서 종목명과 현재가 칸이 좀 커보임, 대비 등락이 좀더 잘 표현되었으면함, 화면에 봉 표현잘안됨, ~내맘대로~ 구성한 화면을 가장먼저 표시되도록 설정저장시 그리 구현되게 등) 담고있는 내용은 매우 좋습니다^^
KB	1	5.2.8	모든 글자좀 크게 할 수 있게 부탁해요 불편해요…?너무
	2	5.2.8	통합검색으로 검색해서 기업 뜨기까지 로딩이 왠말입니까… 로딩 빙글빙글 돌다가 가격 계속 바뀌는데 통합검색 할때 로딩 없애주세요
	3	5.3.6	차트에서 추세선 저장이 잘 될 때도 있고 안될때도 있습니다. 해결책이 있을까요?
	4	5.0.7	위젯기능이 있어서 잘 이용하고있습니다. 다만 가끔 위젯이 먹통이되거나 새로고침, 페이지넘김 반응이 느려지는데 해결 부탁드립니다.
	5	5.2.9	타사 쓰다가 ui가 눈에 잘 들어오고 편해서 완전히 넘어왔어요. 다 좋은데 호가창, 차트 등등 다른 화면 갔다올때 스크롤포지션 유지시켜주면 좋을것 같아요~ 기술적으로 불가능한게 아닌데 기획적인 부분에서 섬세하지 않은 점들이 종종 보이네요ㅠㅠ
Namu	1	8.93	보조지표 종류나 차트 자율도가 타 증권사에 비해 상당히 부족합니다 프로그램 가벼운거 말고는 딱히 장점이 없습니다 부디 키움증권 앱 구경이라도 한번하시고 개선부탁드립니다
	2	8.93	업데이트 후 홈화면 글자가 너무 커요… 한 화면에 한번에 보이는 정보의 수가 줄어들었잖아요 ㅠㅜ
	3	8.93	차트좀 디테일하게… …
	4	8.93	조건검색기능이 좋긴한데 이동평균선 지수로 검색하면 오류가 나서 아무것도 안뜨는것만 해결되면 좋겠습니다.
	5	8.93	정말보기 편하고 사용하기 최고임

**Table 3 behavsci-14-00814-t003:** Examples of the MTS applications’ review data (English translation).

MTS App Name	Score	Review Created Ver.	Review
Korea Investment	1	1.0.16	After the update yesterday, it keeps bugging me and I can’t run the app and make transactions.
	2	1.01.16	When pressing the menu, please maintain the position of the last selected menu. It’s very inconvenient.
	3	1.01.17	Please add the ‘drawer item’ function to the existing chart in the app. This is a very useful feature.
	4	1.01.14	I had a bit of trouble getting used to it, but the stock summary is useful and I am quite satisfied. However, before, I could see the ETF stock discussion room and management fee ratio, but they are gone. Please improve this.
	5	1.0.06	It is easier to use than existing apps. It would be better if a dark mode feature was added.
Mirae Asset	1	7.0.8	I am writing this review after using it for 8 years. The speed is so slow compared to the previous version! The chart is also placed like a banner window, making it less intuitive. Moving between screens is so, so slow!!!!!!
	2	7.0.5	It’s complicated and I like it to be as simple as before, but I can’t see what was done before and it’s very distracting.
	3	7.0.3	The feed continues to appear on the right. No matter how hard I search, it is not hidden, and the screen itself is inconvenient. I was using it really well before, but it was inconvenient after the update.
	4	7.0.5	It’s nice to see a little more progress, it looks cleaner. However, there is a problem with lag when moving to other menus, so I hope it is resolved as soon as possible.
	5	7.0.8	It’s the best~~!! The display is easy to see and seems to have a sophisticated design!! Also, the quote window and order window seem to be user-friendly!
Kiwoom	1	5.5.4	The UI is horribly crude. Not only are menus difficult to find and operate, but even the most basic buy/sell orders are often unintuitive.
	2	5.5.7	I would like to be able to sort the purchased stocks as I wish. It’s very inconvenient because it’s automatically sorted.
	3	5.2.5	I think the function is good, but people who do stock trading on mobile won’t be looking at the screen all the time, so it’s frustrating because when they look at other apps or turn the screen off and on, the Kiwoom app turns off. Is there a setting? It’s like that for both Hero Moon S and Hero Moon G.
	4	5.5.7	It’s cleaner and looks better than the previous app, but it’s a little inconvenient because the highest transaction amount disappears after business hours. I think it would be good to add it.
	5	4.9.5	# It would be good to make a widget # When configuring the mobile screen, I would like to have a more readable screen composition (e.g., the stock name and current column in the items of interest look a bit large, I would like the contrast ups and downs to be expressed better, the bars are not expressed well on the screen, ~ as I like. ~ The content it contains is very good (so that the configured screen is displayed first when saving the settings, etc.) ^^
KB	1	5.2.8	Please make all the letters larger. It’s inconvenient…? Too much.
	2	5.2.8	I searched through integrated search, but what is the loading process until the company pops up? The loading goes round and round, and the price keeps changing. Please remove the loading when using integrated search.
	3	5.3.6	Sometimes trend lines are saved well on charts, and sometimes they are not. Is there a solution?
	4	5.0.7	I use it well because it has a widget function. However, sometimes the widget does not work or the response to refreshing and turning pages is slow. Please solve this problem.
	5	5.2.9	I was using a third-party company, but the UI was easy on the eyes and comfortable, so I completely passed on it. It’s all good, but I think it would be nice to maintain the scroll position when going back and forth between other screens such as the quote window and charts. It’s not technically impossible, but I often see things that aren’t detailed in the planning part.
Namu	1	8.93	The types of auxiliary indicators and chart autonomy are quite lacking compared to other securities companies. There are no particular advantages other than the fact that the program is light. Please take a look at the Kiwoom Securities app and ask for improvements.
	2	8.93	After the update, the text on the home screen is too big… The number of information visible on one screen at once has been reduced.
	3	8.93	A little more detail on the chart… …
	4	8.93	The conditional search function is good, but it would be nice if it could be resolved because when you search using moving average index, an error occurs and nothing is displayed.
	5	8.93	Really easy to look at and great to use

**Table 4 behavsci-14-00814-t004:** The results of categorizing the financial application-review data into five main components.

Aspect									
Update		Display		Authentication		Function		Application	
Component	Count	Component	Count	Component	Count	Component	Count	Component	Count
Version	830	Chart	1421	Sign-in	2377	Account	1904	App	5827
Update	652	Window	1088	Access	2895	Transaction	2151	Apple	3333
Upgrade	208	Smartphone	124	Certificate	1134	Chart	1421	Brokerage	2088
Confirmation	1344	Menu	636	Registration	964	Sell	1089	Stock	2916
Touch	176	Tab	195	Input	573	Buy	1331	Securities	3717
Change	405	Stop	87	Transfer	386	Registration	964	Deletion	962
Sign-in	2377	Screen	2287	Withdrawal	174	Transfer	386	Portfolio	2110
Patch	59	Design	461	Touch	176	Settings	1221		
Update	4402			Authentication	2558				

**Table 5 behavsci-14-00814-t005:** The results of training candidate models for analysis using application-review data.

Model	Accuracy	F1-Score
LSTM	0.79	0.74
1D-CNN	0.69	0.65
Logistic Regression	0.61	0.26
SVM	0.62	0.54

**Table 6 behavsci-14-00814-t006:** The results of review classification using PMI, SVD, and Word2Vec.

Topic	Contents
**Update**	The analysis of reviews related to “update” using PMI, SVD and Word2Vec methodologies revealed numerous issues, such as loading, access, and authentication problems arising after specific updates. PMI and SVD highlighted incompatibility issues with previously stored data, like charts and index settings, while Word2Vec focused on errors occurring post-application, such as updates and the inability to utilize previously available features, mainly reported by existing customers.
**Screen**	The analysis of reviews related to “screen” indicated dissatisfaction with UI screen proportions not matching during device orientation changes and inconvenient button placement leading to unnecessary navigation. PMI and SVD highlighted these layout issues, while Word2Vec focused on the rough UI design and unfavorable comparisons with other securities firms, pinpointing specific discomforting aspects.
**Chart**	Reviews concerning “chart” expressed discomfort with UI navigation, screen layout, and loading speed compared to traditional trading systems while freely navigating between various chart views. PMI and SVD pointed out these navigation and performance issues, whereas Word2Vec addressed dissatisfaction with both UI-related issues like landscape view and functional aspects such as trend lines and additional candlesticks.
**Sign-in**	Reviews regarding “sign-in” addressed login failures due to application errors and frequent automatic logouts while switching between application screens, especially critical during trading hours. PMI and SVD emphasized these frequent logouts and application errors, while Word2Vec focused on server and authentication method errors leading to sign-in failures.
**Access**	Reviews focused on “access” primarily voiced negative sentiments due to connection failures, unresponsiveness, and crashing issues, leading to inconvenience. PMI and SVD highlighted these connection and stability issues, whereas Word2Vec mainly criticized authentication and login errors, emphasizing the importance of seamless application access.
**Authentication**	Reviews concerning “authentication” highlighted problems related to OTP authentication and public key certificate issues, particularly inconvenience during mobile authentication resets or complex authentication procedures. PMI and SVD pointed out these issues, while Word2Vec emphasized errors occurring during various authentication methods such as fingerprint, PIN, and account number verification.
**Account**	The analysis of reviews regarding “account” revealed significant negative sentiments during the account registration process, citing complexities in registration steps like member sign-up, login, and security medium registration, along with authentication problems. PMI and SVD criticized these complex registration processes, while Word2Vec focused on errors during account verification and functional aspects like balance and stock holding checks within the application.
**Transaction**	Reviews concerning “transaction” pointed out difficulties in transactions due to a failure to recognize stored certificates on smartphones and screen blackout during transactions, causing inconvenience, especially critical in time-sensitive stock trading. PMI and SVD highlighted these technical difficulties, while Word2Vec addressed the need for additional functional features during buy/sell transactions and complaints about transaction errors.

## Data Availability

The raw data supporting the conclusions of this article will be made available by the authors on request.

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
