# Peer review of "User Sentiment Analysis Based on Securities Application Elements"

_behavsci, 2024, doi:10.3390/bs14090814_

Round 1

Reviewer 1 Report

Comments and Suggestions for Authors

This paper proposes “User Sentiment Analysis Based on Securities Application Elements “. The subject is worthy of investigation. There are some issues in the paper as follows:

·         The challenges mentioned at the end of the abstract could be expanded upon to provide more insight into the potential limitations of using ABSA for mobile application design research.

·         The mention of techniques such as PMI, SVD, and Word2Vec in the context of analyzing user review data could be elaborated upon to provide a better understanding of the methodology used in the study.

·          

·         There are some errors in the English grammar writing which require careful proof checking and correction.

·         It is better to add their proposed method and your motivations then the amount of improvement in the abstract.

·         It is better to add the dataset in the abstract.

·         It is better to add newer references to the paper

·         What was the reason for using “tf-idf” and wor2vec for embedding over others?

·         In the data Set section, a further explanation of the extracted data should be given and, if possible, examples of that data should be shown.

·         The innovation of your article is not clear

·         discuss your research limitations and complexity of proposed method.

·         future work should be added in the conclusion.

Comments on the Quality of English Language

This paper proposes “User Sentiment Analysis Based on Securities Application Elements “. The subject is worthy of investigation. There are some issues in the paper as follows:

·         The challenges mentioned at the end of the abstract could be expanded upon to provide more insight into the potential limitations of using ABSA for mobile application design research.

·         The mention of techniques such as PMI, SVD, and Word2Vec in the context of analyzing user review data could be elaborated upon to provide a better understanding of the methodology used in the study.

·          

·         There are some errors in the English grammar writing which require careful proof checking and correction.

·         It is better to add their proposed method and your motivations then the amount of improvement in the abstract.

·         It is better to add the dataset in the abstract.

·         It is better to add newer references to the paper

·         What was the reason for using “tf-idf” and wor2vec for embedding over others?

·         In the data Set section, a further explanation of the extracted data should be given and, if possible, examples of that data should be shown.

·         The innovation of your article is not clear

·         discuss your research limitations and complexity of proposed method.

·         future work should be added in the conclusion.

Author Response

Thank you very much for your valuable comments. Below is a summary of how each of your comments has been addressed.

No.

Comments

Response

Lines

1

 The challenges mentioned at the end of the abstract could be expanded upon to provide more insight into the potential limitations of using ABSA for mobile application design research.

Thank you for pointing that out. It seems the abstract did not fully capture the key aspects of the entire paper as clearly as intended. I have revised the abstract to better reflect and concisely summarize the main content and challenges discussed in the study.

10-20

2

 The mention of techniques such as PMI, SVD, and Word2Vec in the context of analyzing user review data could be elaborated upon to provide a better understanding of the methodology used in the study.

Thank you for pointing out the need for further elaboration on the techniques used, such as PMI, SVD, and Word2Vec, in the context of analyzing user review data. In response to your comment, I have added additional details to clarify how these techniques are applied in the study to enhance the readers' understanding of the methodology.

182-190
240-241
257-286
300-301
316-340

3

There are some errors in the English grammar writing which require careful proof checking and correction

I have carefully revised the text and made the necessary corrections to improve the clarity and accuracy of the English grammar. I appreciate your attention to detail and hope the revised version meets the expected standards.

overall scope

4

 It is better to add their proposed method and your motivations then the amount of improvement in the abstract.

Same as Response 1, abstract are revised.

10-20

5

 It is better to add the dataset in the abstract and newer references

As per your comments, I have added brief information of the dataset in Abstract, as well as 18 more recent references.

Dataset : 14-15
Reference[#] : 1-3, 24-26, 28-33, 35-37, 39-41

6

 What was the reason for using “tf-idf” and wor2vec for embedding over others?

 TF-IDF is a frequency-based embedding method that highlights the importance of words within a document through comparison with other documents. This method is useful for identifying and highlighting statistically important words within text. Word2Vec, on the other hand, uses a neural network-based embedding technique to analyze the context in which words appear and reflect the semantic similarity between words in the vector space. This ensures that Word2Vec maintains the contextual meaning of words and understands the semantic relationships of words within the text. Combining the two methods, both the statistical importance and the semantic meaning of the text can contribute to enhancing the accuracy and depth of emotional analysis. Recent models such as BERT and GPT excel in capturing contextual information; however, these approaches are computationally intensive and require substantial resources. Moreover, for research objectives that are relatively straightforward, these advanced models may introduce issues such as overfitting or interpretability challenges.

7

 In the data Set section, a further explanation of the extracted data should be given and, if possible, examples of that data should be shown

As per your comments, I have added explanations and examples regarding the dataset.

358-363

8

 The innovation of your article is not clear

 Korean natural language processing presents unique challenges compared to other languages, and research applying deep learning techniques for sentiment analysis in Korean text remains particularly scarce. In this context, the significance of this paper lies in its attempt to conduct sentiment analysis on Korean reviews, offering valuable insights in this underexplored area. We added this contents in Conclusion section.

542-546

9

 Future work should be added in the conclusion. 

The final paragraph includes the content regarding future work as you suggested.

566-574

Reviewer 2 Report

Comments and Suggestions for Authors

The manuscript investigates the application of ABSA, an interesting problem in the NLP area, with great potential for various business industries. Although methodologically sound, I believe the authors should improve the scope and storytelling. The introduction, for example, gives no details about 'Securities Application,' which appears in the title and as a keyword. In fact, the subject of the reviews analyzed is presented only in section 4.1, when the dataset is introduced. This makes the paper's motivation too shallow, so the research gap must be addressed. To this end, I would recommend a literature update. Since most of the references date back to 2021, there are many new and interesting focuses that the authors could discuss. As a suggestion, observe regional aspects of the language or consumers that might not have been fully explored. Even though we discuss a lot about LLMs nowadays in the NLP area, I believe that language should be regionally discussed, as cultural and local aspects are important when analyzing consumer reviews. I hope these comments can help improve the work

Comments on the Quality of English Language

The English is adequate, with minor errors.

Author Response

Thank you very much for your valuable comments. Below is a summary of how each of your comments has been addressed.

No.

Comments

Response

Lines

1

Although methodologically sound, I believe the authors should improve the scope and storytelling. The introduction, for example, gives no details about 'Securities Application,' which appears in the title and as a keyword.

Thank you for your insightful comment. In response, We have added an explanation in the introduction section detailing why designing securities applications is challenging, particularly highlighting the specific characteristics and context in Korea.

24-47

2

In fact, the subject of the reviews analyzed is presented only in section 4.1, when the dataset is introduced. This makes the paper's motivation too shallow, so the research gap must be addressed. To this end, I would recommend a literature update.

Thank you for your valuable feedback. We have added an explanation of the dataset along with examples in section 4.1. Additionally, as per your suggestion, I have included brief information about the dataset in the abstract to better address the research gap.

14-15

355-363

3

Since most of the references date back to 2021, there are many new and interesting focuses that the authors could discuss.

I have added eight papers published after 2022.

Reference # :

2, 3, 25, 26, 31, 32, 35, 37, 40, 41

4

As a suggestion, observe regional aspects of the language or consumers that might not have been fully explored. Even though we discuss a lot about LLMs nowadays in the NLP area, I believe that language should be regionally discussed, as cultural and local aspects are important when analyzing consumer reviews.

Korean natural language processing presents unique challenges compared to other languages, and research applying deep learning techniques for sentiment analysis in Korean text remains particularly scarce. In this context, the significance of this paper lies in its attempt to conduct sentiment analysis on Korean reviews, offering valuable insights in this underexplored area. We added this contents in Conclusion section.

542-546
